# Short Chain Fatty Acid Acetate Increases TNFα-Induced MCP-1 Production in Monocytic Cells via ACSL1/MAPK/NF-κB Axis

**DOI:** 10.3390/ijms22147683

**Published:** 2021-07-19

**Authors:** Areej Al-Roub, Nadeem Akhter, Amnah Al-Sayyar, Ajit Wilson, Reeby Thomas, Shihab Kochumon, Fatema Al-Rashed, Fahd Al-Mulla, Sardar Sindhu, Rasheed Ahmad

**Affiliations:** 1Immunology & Microbiology Department, Dasman Diabetes Institute, Dasman 15462, Kuwait; areej.abualroub@dasmaninstitute.org (A.A.-R.); nadeem.akhter@dasmaninstitute.org (N.A.); amnah.alsayyar@dasmaninstitute.org (A.A.-S.); ajit.wilson@dasmaninstitute.org (A.W.); reeby.thomas@dasmaninstitute.org (R.T.); shihab.kochumon@dasmaninstitute.org (S.K.); fatema.alrashed@dasmaninstitute.org (F.A.-R.); 2Genetics & Bioinformatics, Dasman Diabetes Institute, Dasman 15462, Kuwait; fahd.almulla@dasmaninstitute.org; 3Animal and Imaging Core Facilities, Dasman Diabetes Institute, Dasman 15462, Kuwait; sardar.sindhu@dasmaninstitute.org

**Keywords:** short-chain fatty acids, acetate, TNFα, THP-1 monocytic cells, MCP-1, NF-κB, AP-1

## Abstract

Short-chain fatty acid (SCFA) acetate, a byproduct of dietary fiber metabolism by gut bacteria, has multiple immunomodulatory functions. The anti-inflammatory role of acetate is well documented; however, its effect on monocyte chemoattractant protein-1 (MCP-1) production is unknown. Similarly, the comparative effect of SCFA on MCP-1 expression in monocytes and macrophages remains unclear. We investigated whether acetate modulates TNFα-mediated MCP-1/CCL2 production in monocytes/macrophages and, if so, by which mechanism(s). Monocytic cells were exposed to acetate with/without TNFα for 24 h, and MCP-1 expression was measured. Monocytes treated with acetate in combination with TNFα resulted in significantly greater MCP-1 production compared to TNFα treatment alone, indicating a synergistic effect. On the contrary, treatment with acetate in combination with TNFα suppressed MCP-1 production in macrophages. The synergistic upregulation of MCP-1 was mediated through the activation of long-chain fatty acyl-CoA synthetase 1 (ACSL1). However, the inhibition of other bioactive lipid enzymes [carnitine palmitoyltransferase I (CPT I) or serine palmitoyltransferase (SPT)] did not affect this synergy. Moreover, MCP-1 expression was significantly reduced by the inhibition of p38 MAPK, ERK1/2, and NF-κB signaling. The inhibition of ACSL1 attenuated the acetate/TNFα-mediated phosphorylation of p38 MAPK, ERK1/2, and NF-κB. Increased NF-κB/AP-1 activity, resulting from acetate/TNFα co-stimulation, was decreased by ACSL1 inhibition. In conclusion, this study demonstrates the proinflammatory effects of acetate on TNF-α-mediated MCP-1 production via the ACSL1/MAPK/NF-κB axis in monocytic cells, while a paradoxical effect was observed in THP-1-derived macrophages.

## 1. Introduction

Obesity is a complex chronic disease that afflicts more than one in two adults and one in six children worldwide. Current data suggest that obesity mediated low-grade chronic inflammation plays a pivotal role in obesity related disorders [1,2]. One of the most common metabolic diseases that is highly linked to obesity is type 2 diabetes (T2DM). The nature of the diabetes–obesity relationship remains controversial because not all obese individuals develop diabetes. However, it is believed that insulin resistance and insulin deficiency are the primary factors that are closely related to the insulin secretion cycle in obesity [3,4].

Adipose tissue is a complex and highly active metabolic and endocrine organ that plays a major role in coordinating a variety of biological processes such as energy metabolism as well as neuroendocrine and immune function [5]. In obesity, adipose tissue is dysregulated, especially in the visceral compartment, and is known to be a significant risk factor for insulin resistance [6]. Such alterations in the adipose tissue lead to adipocyte hypertrophy, immune cell infiltration, and increased cytokine (IL-1β, IL-6, TNFα) and chemokine production, which eventually lead to insulin resistance, T2DM, dyslipidemia, and hypertension [7]. Insulin resistance is also characterized by the circulation of high levels of free fatty acids (FFAs), which induce glucose uptake in the liver and muscle [2]. FFAs are considered an energy source for most tissues and play a critical role in gene expression and enzymatic regulation [8]. FFAs are either derived from dietary intake or are inherently produced by the lipolysis of stored triglycerides. They are known to play a major role in metabolic inflammation.

Short-chain fatty acids (SCFAs) are formed as products of the fermentation of dietary carbohydrates, peptides, and proteins in the gut microbiota. They perform several physiological roles in regulating intestinal permeability, inflammation control, immunological function, and serve as an energy source for host colonocytes [9]. The most abundant SCFAs are acetate, propionate, and butyrate with a molar ratio of 60:20:20, respectively. Despite their low concentration, propionate and butyrate have a distinct role in reducing inflammation in the gut as well as regulating gene expression and cell fate [9]. On the other hand, acetate, which is the most abundant SCFA, exhibits a number of physiological functions from acting as a substrate for cholesterol synthesis to functioning as an appetite suppressor in the hypothalamus [10]. Acetate is well known for its anti-inflammatory function [11,12]. Acetate and butyrate play a significant role in modulating the production of chemokines and the expression of neutrophils and endothelial cells. The inhibitory effects of SCFAs on MCP-1 and LPS-induced IL-10 production have been reported [13,14].

Obesity is associated with an increase in adipose tissue size and results in the activation of macrophage inflammatory factors including TNFα, IL-6, and MCP-1 (CCL2), which causes insulin resistance [5]. CCL2 and chemokine (C-C motif) receptor 2 (CCR2) knockout mice exhibit reduced adipose tissue macrophages, lower inflammatory levels, and enhanced insulin sensitivity with a high-fat-diet (HFD) [15,16]. In addition, Takahashi et al. suggested that increased circulation of MCP-1 in obese mice is associated with an increase in monocyte/macrophage circulation in the blood [17]. TNFα increases the production of MCP-1 during conditions of obesity. Since SCFAs are involved in the suppression of inflammatory responses, we investigated the effects of SCFAs on the TNFα -mediated production of MCP-1 in monocytic cells. Our findings show that acetate in combination with TNFα synergize MCP-1 production from monocytic cells through the ACSL1/MAPK/NF-κB axis.

## 2. Results

### 2.1. TNFα Enhances MCP-1 Production by Monocytic Cells in the Presence of Acetate

We investigated whether acetate modulates TNFα-induced MCP-1 production in monocytic cells. Monocytic cells were treated with TNFα in the presence or absence of acetate for 24 h. The results indicated that MCP-1 gene/protein expression in THP-1 monocytic cells was significantly upregulated after co-treatment with TNFα and acetate compared to TNFα treatment alone (MCP-1 mRNA: *p* = 0.0051; MCP-1 protein: *p* < 0.0001) (Figure 1A,B). Western blot and confocal microscopy of the monocytic cells also revealed that there was a significant increase in the expression of MCP-1 in the cells treated with acetate and TNFα (Figure 1C–F).

Next, to determine whether synergy with TNFα is specific to acetate, we treated monocytic cells with TNFα and the other SCFAs, butyrate, and propionate. We found that butyrate and propionate also synergized with TNFα to induce the production of MCP-1 (Figure 2A,B).

Moreover, we compared MCP-1 production in monocytes induced through co-stimulation with TNFα and acetate versus TNFα and LPS (TLR4 agonist). The data show that TNFα/LPS induced significantly higher MCP-1 production compared to induction through TNFα/acetate (Appendix A).

Macrophages are key contributors to metabolic inflammation [18]. To determine whether this cooperative effect of acetate/TNFα was reproducible in macrophages, we treated macrophages with TNFα in the presence or absence of acetate, butyrate, or propionate. Interestingly, our data showed that TNFα mediated MCP-1 production was significantly downregulated in macrophages pretreated with acetate (Figure 3A) (*p* < 0.0001), butyrate (Figure 3B) (*p* < 0.0001), or propionate (Figure 3C) (*p* < 0.0001).

### 2.2. Inhibition of ACSL1 Suppresses the Synergistic Production of MCP-1 by Acetate/TNFα

ACSL1 is a key enzyme of lipid/fatty acid metabolism and is involved in the TNFα-induced immune response of monocytic cells [19]. To determine whether ACSL1 is required for the acetate/TNFα-mediated activation of monocytes for the production of MCP-1, we used triacsin c to inhibit the ACSL1 in monocytes. The results showed that pretreatment of the monocytes with triacsin c followed by exposure to acetate/TNFα caused a significant downregulation of MCP-1 expression (MCP-1 mRNA: *p* < 0.0001; MCP-1 protein: *p* = 0.0003) (Figure 4A,B). However, no differences were observed for the suppression of MCP-1 when the cells were pretreated with the lipid metabolism inhibitors etomoxir (carnitine palmitoyltransferase 1 (CPT1)) or myriocin (serine palmitoyltransferase (SPT)) (Figure 4C,D).

TLR signaling pathways are required for the production of particular cytokines during obesity. The specific receptor is most likely engaged by SCFAs to induce the inflammatory response [20]. All TLRs expressed on the cell surface utilize MyD88 for signal transduction upon ligand binding [21]. To determine the role of MyD88 in the synergistic production of MCP-1 by acetate/TNFα, monocytic cells deficient in MyD88 were used. We found that MCP-1 expression in response to the acetate/TNFα co-stimulation of the MyD88-defective cells was not affected (Appendix A). Furthermore, acetate/TNFα-mediated NF-κB/AP-1 activity was not suppressed in the MyD88-deficient cells (*p* < 0.0001) (Appendix A), suggesting that there was no role for the TLRs in the synergistic effect. Altogether, these data indicate that the cooperative induction of MCP-1 with acetate/TNFα does not require TLRs and is significantly dependent on ACSL1.

### 2.3. ACSL1 Deficiency Inhibits Acetate/TNFα-Mediated the Synergistic Production of MCP-1

To further verify acetate/TNFα-mediated synergy for MCP-1 production in monocytes is dependent on ACSL1, we transfected cells with ACSL1 siRNA. This achieved a reduction in the ACSL1 mRNA levels that was greater than 50% compared to scrambled (control) siRNA (Neg-siRNA) (ACSL-1 mRNA: *p* = 0.0182) (Figure 5A). Accordingly, the expression of MCP-1 mRNA/protein was significantly reduced in the ACSL1 siRNA-transfected cells after stimulation with acetate /TNF-α compared to the scrambled siRNA-transfected cells (MCP-1 mRNA: *p* < 0.0001; MCP-1 protein: *p* = 0.0150) (Figure 5B,C).

### 2.4. Synergistic MCP-1 Expression by Acetate/TNFα Involves MAPK/NF-κB Signaling Pathways

The downstream signaling of SCFAs and TNFα may lead to the activation of the MAPK and NF-κB pathways [22,23,24,25]. Next, we investigated whether the cooperative production of MCP-1 by acetate/TNFα co-stimulation involved signaling through the MAPK and/or NF-κB pathways. The results showed that the combined effects of acetate/TNFα on MCP-1 expression at both the mRNA and protein levels were significantly decreased in monocytic cells when they were preincubated with MAPK (JNK, ERK1/2, MEK) pathway inhibitors, including SP600125 (mRNA: *p* < 0.0001; protein: *p* < 0.0001), PD98059 (mRNA: *p* < 0.0001; protein: *p* < 0.0001), SB203580 (mRNA: *p* < 0.0001; protein: *p* < 0.0001), U0126 (mRNA: *p* < 0.0001; protein *p* < 0.0001), and XMD8-92 (mRNA: *p* < 0.0001; protein: *p* < 0.0001) (Figure 6A,B). MCP-1 mRNA/protein expression induced by acetate/TNFα was suppressed in cells pretreated with NF-κB pathway inhibitors, including NDGA (mRNA: *p* = 0.0015; protein: *p* < 0.0001), Bay11-7085 (mRNA: *p* < 0.0001; protein: *p* < 0.0001), trolox (mRNA: *p* = 0.0017; protein: *p* < 0.0001), triptolide (mRNA: *p* < 0.0001; protein: *p* < 0.0001), and resveratrol (mRNA: *p* < 0.0001; protein: *p* < 0.0001) (Figure 6C,D).

### 2.5. Involvement of ACSL1 in the Acetate/TNFα Mediated MAPK/NF-κB Phosphorylation

The data showed that the inhibition of MAPK and NF-*κ*B suppressed the synergistic effect of acetate/TNFα on the production of MCP-1. Moreover, long-chain acyl-CoA has been shown to induce inflammation through MAPK/NF-κB activation [19,26,27]. Therefore, we determined whether ACSL1 had an impact on the acetate/TNFα-mediated phosphorylation of JNK, P38, ERK, and NF-κB. Monocytic cell treatment with acetate/TNFα increased the phosphorylation of JNK, P38, ERK, and NF-κB, which was decreased when the cells were preincubated with an ACSL1 inhibitor (triacsin c) (Figure 7A,B).

### 2.6. Involvement of NF-κB and AP-1 in Acetate/TNFα-Induced MCP-1 Production

We also used NF-κB/AP-1 reporter cells that had been stably transfected with a secreted embryonic alkaline phosphatase (SEAP) reporter construct linked to the NF-κB/AP-1 promoter. SEAP reporter activity was significantly upregulated when cells were exposed to acetate/TNFα compared to TNFα alone (*p* < 0.0001) (Figure 8A). Acetate/TNFα-mediated NF-κB/AP-1 activation was significantly suppressed when the cells were pretreated with triacsin c (*p* < 0.0001) (Figure 8B). Similarly, the data showed that acetate/TNFα synergistically upregulated MCP-1 production in NF-κB/AP-1 reporter monocytic cells (mRNA: *p* < 0.0001; protein: *p* < 0.0001) (Figure 8C,D). However, the inhibition of other lipid metabolism enzymes (e.g., CPT-1 with etomoxir or SPT with myriocin) did not show any effect on acetate/TNFα-induced NF-κB/AP-1 activation (Figure 8E,F).

To summarize the underlying signaling pathway involved in this cooperative relationship between acetate and TNF-α for MCP-1 production, a schematic illustration is presented in Figure 9.

## 3. Discussion

Dysfunctional white adipose tissue increases the risk of insulin resistance in obesity. Adipose tissues develop chronic low-grade inflammation during conditions of obesity [1]. Inflammation in adipose tissue is characterized by the overproduction of cytokines/chemokines and an increase in the number of macrophages [2,28]. MCP-1 and CCL2 function by activating monocytes to leave circulation and develop into adipose tissue macrophages, the first phase in the initiation of adipose inflammation [28]. In obese mice and humans, MCP-1 and TNFα production are increased in both plasma and adipose tissue [17,29]. TNFα robustly enhances the production of chemokines such as MCP-1 and IL-8 by monocytic cells under the influence of long chain fatty acids [28,30]. Although, SCFAs exhibit anti-inflammatory properties and may potentially be used for the treatment of inflammatory diseases [13], the direct effect of SCFAs on TNFα-mediated MCP-1 production by monocytic cells remains unclear. Here, we report for the first time that TNFα significantly increases MCP-1 production in monocytic cells in the presence of SCFAs. More interestingly, we found that in contrast to monocytic cells, SCFAs suppressed the TNFα-induced MCP-1 production in the macrophages, showing a differential role of SCFAs related to monocytes and macrophages. In general, SCFAs inhibit LPS-mediated production of inflammatory cytokines [13,14]. We observed similar effects in the macrophages. SCFAs enhance the inflammatory response by activating TLRs [20]. Interestingly, lower levels of SCFAs significantly enhance the TLR2 ligand and the TLR7 ligand-induced production of IL-8 and TNFα in a time- and dose-dependent manner, but it has little effect on lipopolysaccharide-induced cytokine release [31]. We found that the TNFα/LPS induced production of MCP-1 was relatively higher than that induced by TNFα/acetate (Appendix A). Our data also show that combined with TNFα, an SCFA such as acetate has a paradoxical effect regarding MCP-1 production in monocytes (MCP-1 upregulation) and macrophages (MCP-1 downmodulation). These studies collectively support that SCFAs can exert both anti-inflammatory and proinflammatory responses depending on concentration, cell type, and culture conditions [23].

Understanding the role of TLR signaling with respect to a particular cytokine in obesity will provide insight into how SCFAs induce inflammatory responses [14,31]. All surface TLRs utilize the MyD88 adaptor protein for signal transduction upon ligand binding [21]. Therefore, it is important to determine whether the synergistic upregulation of MCP-1 by acetate/TNFα implicates MyD88-dependent mechanisms. We found that the synergistic production of MCP-1 induced by acetate/TNFα was not affected in monocytic cells deficient in MyD88. These results clearly show that there is no significant TLR involvement as SCFA signaling sensors cooperating with TNFα for MCP-1 expression in monocytic cells. However, it is interesting to note that mammalian G protein-coupled receptors FFAR2 (also called GPR43) and FFAR3 (also called GPR41) can also act as receptors for SCFAs. These receptors are expressed on both human and mouse monocytes and may modulate inflammatory responses in response to SCFAs. The treatment of human monocytes with acetate, SCFA, or FFAR2-/FFAR3-specific synthetic agonists induced elevated p38 phosphorylation and reduced the expression of the IL-1α, IL-1β, C5, CCL1, CCL2, GM-CSF, and ICAM-1 inflammatory cytokines. Ang et al. reported that human and mouse monocytes showed differential signaling and cytokine profiles following stimulation with synthetic agonists of FFAR2 and FFAR3 and that the acetate-induced elevation of IL-1α/β and GM-CSF persisted in FFAR2/3 knockout mice but could not be reproduced using synthetic agonists, suggesting a FFAR2/3-independent mechanism in mice [32].

ACSL1 is a key enzyme in lipid/fatty acid metabolism and is involved in the TNFα-mediated proinflammatory phenotypic shift in monocytes [19,26]. Our data show that blocking the activity of ACSL1 with triacsin c treatment inhibits the synergistic production of MCP-1 by acetate/TNFα. ACSL1 deficiency is involved in preventing the TNFα-induced inflammatory switch in monocytes, supporting the role of ACSL1 in inflammation [19,33]. Inhibiting ACSL1 in monocytes suppressed the TNF-α-induced expression of CD11c, which is a highly expressed inflammatory marker on monocytes and macrophages during conditions of obesity [19]. ACSL1 regulates the TNFα- and LPS-induced GM-CSF production in breast cancer MDA-MB-231 cells [26,27], suggesting the role of ACSL1 in the regulation of growth-modulating cytokines or embryokines. Increased ACSL1 expression was detected in inflammatory macrophages from obese mice and humans [33]. ACSL1 is upregulated by LPS and TNFα, both of which are elevated in obesity [33]. ACSL1 expression is high in TNFα-activated inflammatory monocytes. Similarly, Al-Rashed et al. showed that the disruption of ACSL1, an enzyme responsible for the esterification of saturated fatty acids, reduces phenotypic inflammatory marker (CD16, CD11b, CD11c and HLA-DR) expression in monocytic cells, thereby decreasing IL-1β and MCP-1 release [19]. This suggests the importance of ACSL1 in the regulation of TNFα-induced monocyte activation. In addition, the myeloid cell-specific deletion of ACSL1 in a diabetic mouse model prevented the formation of the inflammatory macrophage phenotype and development of diabetes [33]. Because ACSL1 is involved in the proinflammatory response induced by TNFα, this may explain why the disruption of ACSL1 reduces the acetate/TNFα induced synergistic production of MCP-1 in monocytic cells. We further investigated the association of ACSL1 with different pathways involved in lipid metabolism, including CPT1 and SPT, but no significant impact was observed. Indeed, fatty acids may be metabolized through several pathways including β-oxidation, triglyceride synthesis, phospholipid synthesis, cholesterol ester synthesis, fatty acid elongation, and protein acylation and may also serve as signaling molecules. Since ACSL1, CPT1, and SPT are the enzymes shown to play essential roles in lipid metabolism [34,35,36], we chose to assess their role in the TNFα/acetate-induced MCP-1 production by monocytic cells.

To further understand the downstream effects of acetate/TNFα stimulation of monocytic cells, we examined the NF-κB, p38 MAPK, ERK, and JNK signaling pathways, all of which have been shown to stimulate MCP-1 production [28]. TNFα/acetate-mediated MCP-1 production was significantly reduced by the inhibition of p38 MAPK, ERK1/2, JNK, and NF-κB signaling. It is well documented that TNFα stimulates the MAPK and NF-κB signaling pathways involved in the regulation of several inflammatory cytokines that contribute to the pathogenesis of various inflammatory conditions [37,38]. It has been reported that TNFα-induced GM-CSF secretion by human lung fibroblasts was partially blocked by the inhibitors of ERK and p38 MAPK [39]. Consistent with previous findings in different cells, we found that acetate/TNFα induced the production of MCP-1 by monocytic cells, which was significantly blocked by inhibiting p38 MAPK and ERK1/2 signaling. This suggests that MCP-1 production in response to TNFα is regulated in different cell lines similarly. More importantly, we found that the activation of p38 MAPK, JNK, ERK1/2, and NF-κB is the underlying mechanism for the overproduction of MCP-1 since the increased phosphorylation of MAPK, JNK, ERK1/2, and NF-κB was detected in cells co-stimulated with acetate/TNFα compared to those stimulated by TNFα alone.

Interestingly, our results also show that blocking ACSL1 activity inhibited the acetate/TNFα-induced phosphorylation of p38 MAPK, ERK1/2, JNK, and NF-κB in monocytic cells. ACSL1 genetic silencing also reduced the TNFα-mediated phosphorylation of p38 MAPK, ERK1/2, JNK, and NF-κB. These results suggest that ACSL1 acts upstream of the MAPK and NF-κB signaling pathways. NF-κB and AP-1 pathways are involved in the regulation of various proinflammatory genes, including those encoding cytokines/chemokines.

In conclusion, our data reveal a novel role of ACSL1 in the acetate/TNFα-mediated production of MCP-1 by monocytic cells that depends, at least in part, on the activation of the MAPK and NF-κB signaling pathways. These findings also point to a likely association between acetate levels, TNFα expression, and ACSL1 activity in obesity that may contribute to metabolic inflammation.

## 4. Materials and Methods

### 4.1. Cell Culture and Stimulation

The human THP-1 monocytic cell line (monocytic cells) was obtained from the American Type Culture Collection (ATCC; Manassas, VA, USA). The cells were cultured in RPMI-1640 complete medium (Gibco, Life Technologies, Grand Island, NY, USA) containing 10% FBS (Gibco, Life Technologies, Grand Island, NY, USA), 2 mM glutamine (Gibco, Life Technologies, Grand Island, NY, USA), 1 mM sodium pyruvate, 10 mM HEPES, 50 U/mL penicillin, 50 μg/mL streptomycin, and 100 μg/mL Normocin, (Gibco, Life Technologies, Grand Island, NY, USA) and incubated at 37 °C with humidity and 5% CO_2_. THP-1-XBlue cells stably expressing the NF-κB/AP-1 inducible SEAP reporter as well as the THP-1 cells deficient in MyD88 activity, known as THP-1-XBlue™-defMyD or MyD88−/− THP-1, were purchased from a commercial source (InvivoGen, San Diego, CA, USA) and cultured as described earlier [40,41]. First, THP1-XBlue cells were cultured in RPMI-1640 complete medium containing Zeocin (200 μg/mL; InvivoGen, San Diego, CA, USA) to maintain stable expression of the NF-κB/AP-1 driven SEAP reporter. THP-1-XBlue™-defMyD cells were cultured in RPMI-1640 complete medium containing Zeocin (200 μg/mL), Normocin (200 ug/mL), and HygroGold (100 μg/mL; InvivoGen, San Diego, CA, USA) [28]. Before stimulation, THP-1 cells were transferred to normal medium and plated in 12-well plates (Costar, Corning Incorporated, Corning, NY, USA) at a cell density of 1 × 10^6^ cells per well unless otherwise stated. Cells were stimulated with sodium acetate (100 mM; Sigma, Saint Louis, MO, USA), sodium butyrate (2 mM; Sigma, Saint Louis, MO, USA), sodium propionate (10 mM; Sigma, Saint Louis, MO, USA), and/or TNFα (10 ng/mL; Sigma, Saint Louis, MO, USA) or 0.1% BSA (Sigma, San Diego, CA, USA) and incubated at 37 °C for 24 h. In some experiments, monocytic cells were stimulated with TNFα (10 ng/mL) and/or LPS (10 ng/mL) and incubated at 37 °C for 24 h. Cells were harvested for RNA isolation and culture supernatants were analyzed for MCP-1 measurement.

### 4.2. Macrophage Differentiation

Macrophages derived from monocytes as described earlier [42] were used for different treatments. THP-1 cells (1 × 10^6^ cells per mL) were cultured using 12-well plates (Costar, Corning Incorporated, Corning, NY, USA) in RPMI 1640 medium (Life Technologies, Grand Island, NY, USA) containing 10% FBS (Life Technologies), 2 mM glutamine, 1 mM sodium pyruvate, 10 mM HEPES, 100 μg/mL Normocin, 50 U/mL penicillin, and 50 μg/mL streptomycin and incubated at 37 °C in 5% CO_2_ under humidity. THP-1 cells were differentiated into macrophages using PMA treatment (10 ng/mL) for 3 d in routine culturing media. Cells were washed and then incubated in RPMI (10% FBS) for 48 h before various treatments [43].

### 4.3. Real-Time RT-PCR

Total cellular RNA was extracted using the RNeasy Mini Kit (Qiagen, Valencia, CA, USA) following the manufacturer’s instructions [44]. Complementary DNA (cDNA) was synthesized using 1 μg of total RNA following the guidelines from the high-capacity cDNA reverse transcription kit (Applied Biosystems, Foster City, CA, USA). For each real-time PCR reaction, 50 ng of cDNA template was amplified using Inventoried TaqMan Gene Expression Assay products (MCP-1: Hs00234140_m1; ACSL-1: Hs00960572_g1, GAPDH: 4310884E) using two gene-specific primers, one TaqMan MGB probe (6-FAM dye-labeled), TaqMan^®^ Gene Expression Master Mix (Applied Biosystems, Foster City, CA, USA), and a 7500 Fast Real-Time PCR System (Applied Biosystems, Foster City, CA, USA) [28]. The target mRNA levels were normalized against GAPDH mRNA relative to the control and calculated using the 2^−ΔΔCT^ method [45,46,47,48]. Relative mRNA expression was expressed as fold expression relative to the average of control gene expression. The expression level in the controls was designated as 1 [49,50].

### 4.4. ELISA

Secreted MCP-1 protein levels were measured in supernatants of acetate and/or TNFα stimulated THP-1 monocytic cells using sandwich ELISA following the manufacturer’s instructions (DuoSet, DY279, Minneapolis, MN, USA) [28].

### 4.5. Small Interfering RNA (siRNA) Transfection

Monocytes were washed and resuspended in 100 uL of the nucleofector solution provided with the Amaxa Noclecfector Kit V, transfected separately with siRNA-ACSL1 (30 nM; OriGene Technologies, Inc., Rockville, MD, USA), and scrambled (control) siRNA (30 nM; OriGene Technologies, Inc., Rockville, MD, USA). All transfection experiments were performed with the Amaxa Cell Line Nucleofector Kit V for monocytic cells (Lonza, Cologne, Germany) using the Amaxa Electroporation System (Amaxa Inc., Cologne, Germany) according to the manufacturer’s protocol [47,51]. After 36 h of transfection, the cells were treated with TNF-α and acetate for 24 h. Cells were harvested for RNA isolation, and the supernatants were collected for MCP-1 protein analysis. ACSL1 gene knockdown levels were assessed with real-time PCR using ACSL1 gene-specific primers/probes.

### 4.6. Immunocytofluorescence

Immunofluorescence was performed as described previously [26]. First, following treatment, 1 × 10^6^ monocytic cells were washed with PBS and coated on slides using a cytospin technique at 600 rpm for 3 min. The slides were fixed in 4% formaldehyde and washed three times with cold PBS. Cells were permeabilized with 0.1% Triton X-100, washed three times in cold PBS, blocked in 1% BSA for 1 h, and incubated overnight with primary Ab (1:200 rabbit anti-human MCP-1 polyclonal Ab; ab9669; Abcam) at room temperature. Cells were washed three times in PBS with 0.05% Tween and incubated with secondary Ab (Alexa Fluor 488 conjugated; ab150077; Abcam) for 1 h. After several washes with PBS, cells were counterstained and mounted using VECTASHIELD HardSet Antifade Mounting Medium with DAPI (catalog number H-1500; Vector Laboratories). Confocal images were collected using a Plan-Apochromat 63×/1.40 oil DIC M27 objective lens (Inverted Zeiss LSM 710 Axio Observer microscope; Gottingen, Germany) with excitation via a 590-nm diode-pumped solid-state laser and a 405-nm line of an argon ion laser, and the optimized emission detection bandwidths were configured using Zeiss ZEN 2010 software [28].

### 4.7. Western Blot Analysis

Following different treatments, monocytic cells were harvested and incubated for 30 min with lysis buffer (10× Lysis Buffer, Cell Signaling Technology Inc., Danvers, MA, USA). The protein lysates were prepared and resolved using 12% SDS-PAGE as described earlier [44,52]. Cellular proteins were transferred to Immuno-Blot PVDF membranes (Bio-Rad Laboratories, Hercules, CA, USA) by electro blotting. The membranes were then blocked with 5% non-fat milk in PBS for 1 h followed by incubation with primary antibodies against p-JNK and JNK, p-P38 and P38, p-ERK and ERK, and p-NF-κB and NF-κB at a 1:1000 dilution at 4 °C overnight. All the primary antibodies were purchased from Cell Signaling (Cell Signaling Technology Inc., Danvers, MA, USA). Anti-MCP1 antibody was bought from abcam (abcam, Cambridge, MA, USA), and B-actin was purchased from Cell Signaling Technology, Inc. The blots were then washed three times with TBS-T and incubated for 2 h with HRP-conjugated secondary antibody (Promega, Madison, WI, USA). Immunoreactive bands were developed using an Amersham ECL Plus Western Blotting Detection System (GE Health Care, Buckinghamshire, UK) and visualized with a Molecular Imager^®^ (VersaDocTM MP Imaging Systems, Bio-Rad Laboratories, Hercules, CA, USA) [53].

### 4.8. Measurement of NF-κB/AP-1 Activity

NF-κB/AP-1 activity reporter monocytic cells (THP1-XBlue cells; InvivoGen, San Diego, CA, USA) were stably transfected with a reporter construct expressing the SEAP gene under control of a promoter induced by the transcription factors NF-κB and AP-1. The cell stimulation resulted in NF-κB and AP-1 activation and SEAP expression. Reporter cells were stimulated with acetate (100 mM) and/or TNFα (10 ng/mL) or 0.1% BSA and cell cultures were incubated at 37 °C for 24 h. SEAP levels were measured by incubating culture supernatants for 3 h with QUANTI-BlueTM solution (InvivoGen, San Diego, CA, USA) and measuring the absorption at 650 nm [28].

### 4.9. Statistical Analysis

Statistical analysis was performed using GraphPad Prism software (La Jolla, CA, USA). Data are shown as mean ± standard error of the mean, unless otherwise indicated. Unpaired Student *t*-tests and one-way ANOVA followed by Tukey’s test were used to compare means between groups. For all analyses, data from a minimum of three sample sets were used for statistical calculation. *p* value < 0.05 was considered significant. Ns: no significance, * *p* < 0.05, ** *p* < 0.01, *** *p* < 0.001 and **** *p* < 0.0001).

## Figures and Tables

**Figure 1 ijms-22-07683-f001:**
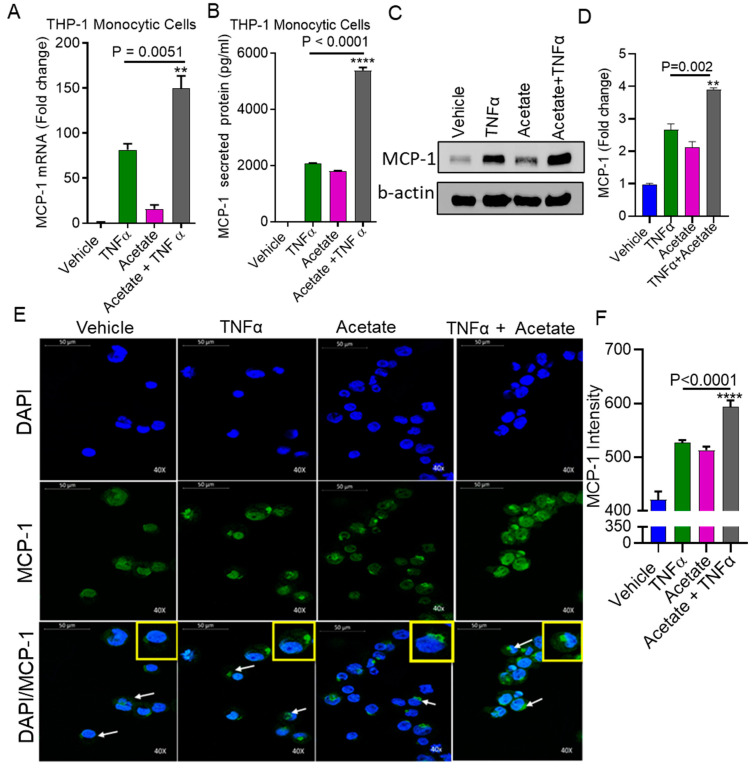
Acetate triggers TNFα induced MCP-1 expression in human monocytic cells. THP-1 monocytic cells were stimulated with acetate (100 mM) and TNFα (10 ng/mL) alone or in combination for 24 h. Cells and culture media were collected. (**A**) Total RNA was extracted from the cells and MCP-1 mRNA was quantified by real time PCR. Relative mRNA expression was expressed as a fold change. (**B**) Secreted MCP-1 protein in culture media was determined by ELISA. (**C**) Cells were treated as described earlier. 8 h before harvesting, cells were incubated with 1 uL/mL of Brefeldin A (Brefeldin A Solution (1000×); Invitrogen, Thermo Fisher Scientific, Waltham, MA, USA). MCP-1 was determined by Western blot. (**D**) Quantification of Western blot. (**E**) THP-1 cells were immune-stained for confocal microscopy, as described in Materials and Methods. MCP-1 expression is shown by green fluorescence (inset), whereas nuclei are stained blue with DAPI (original magnification × 40). (**F**) MCP-1 fluorescence intensity was determined for 10 random images. The results obtained from three independent experiments are shown. All data are expressed as mean ± SEM (n = 3). ** *p* < 0.01, **** *p* < 0.0001 versus TNFα alone.

**Figure 2 ijms-22-07683-f002:**
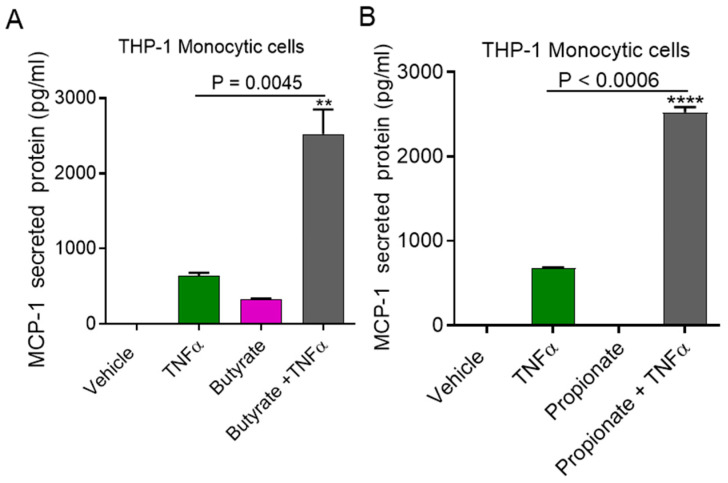
Impact of butyrate and propionate on TNFα induced MCP-1 expression. (**A**,**B**) THP-1 monocytic cells were stimulated with butyrate (2 mM), propionate (10 mM), and TNFα (10 ng/mL) alone or in combination for 24 h. Secreted MCP-1 protein in culture media was determined by ELISA. The results obtained from three independent experiments are shown. All data are expressed as mean ± SEM (n = 3). ** *p* < 0.01, **** *p* < 0.0001 versus TNFα alone.

**Figure 3 ijms-22-07683-f003:**
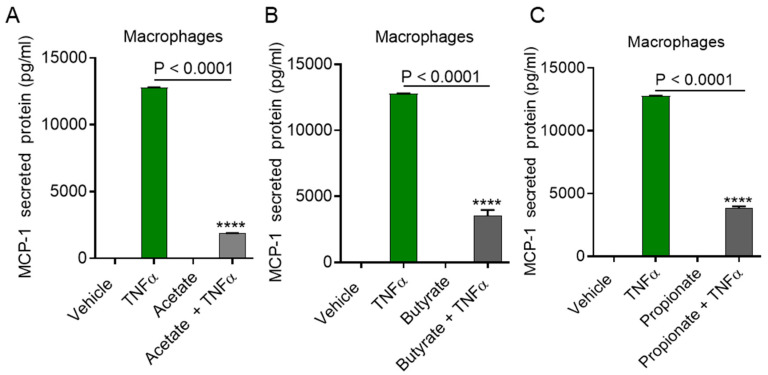
Impact of acetate on TNFα induced MCP-1 expression in macrophages. Monocytic cells derived macrophages were stimulated with acetate (100 mM), butyrate (2 mM), propionate (10 mM), and TNFα (10 ng/mL) alone or in combination for 24 h. (**A**–**C**) Secreted MCP-1 protein in culture media was determined by ELISA. The results obtained from three independent experiments are shown. All data are expressed as mean ± SEM (n = 3). **** *p* < 0.0001 versus TNFα alone.

**Figure 4 ijms-22-07683-f004:**
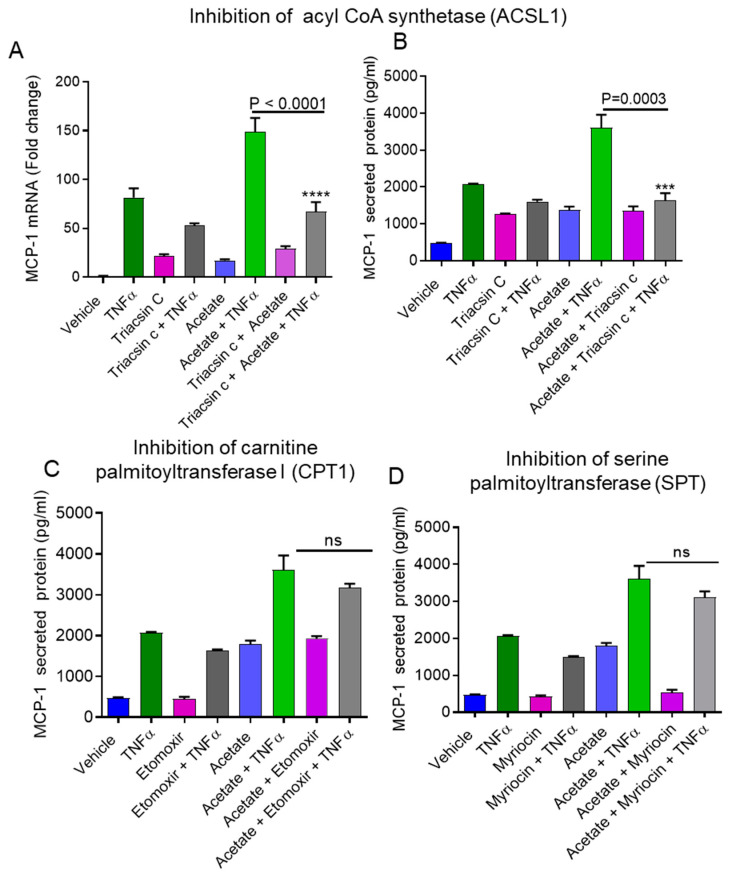
ACSL1 inhibition reduces acetate/TNFα mediated synergistic MCP-1 production in monocytes. Monocytic cells were pretreated with inhibitors (Triacsin c:(4 uM), etomoxir (10 uM), myriocin (50 nM)) or vehicle for 1 h and then incubated with acetate/TNFα for 24 h. (**A**) MCP-1 mRNA was determined by real-time PCR and (**B**–**D**) MCP-1 protein was determined by ELISA. The results obtained from three independent experiments are shown. All data are expressed as mean ± SEM (n = 3). *** *p* < 0.001, **** *p* < 0.0001 versus TNFα alone. ns indicates non-significant.

**Figure 5 ijms-22-07683-f005:**
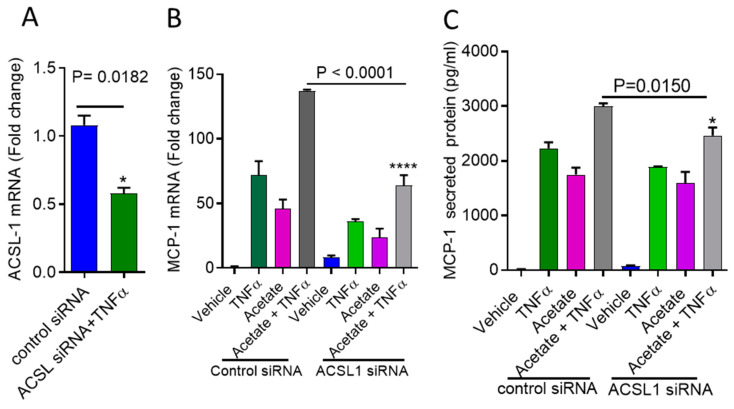
Acetate synergy with TNFα for MCP-1 production requires ACSL1. (**A**) THP-1 monocytic cells were transfected with either control or ACSL1 siRNA and incubated for 36 h. Real time PCR was done to measure ACSL1 expression. (**B**,**C**) ACSL1 deficient THP-1 cells were stimulated with acetate and TNFα. MCP-1 expression was determined. The results obtained from three independent experiments are shown. All data are expressed as mean ± SEM (n = 3). * *p* < 0.05; **** *p* < 0.0001.

**Figure 6 ijms-22-07683-f006:**
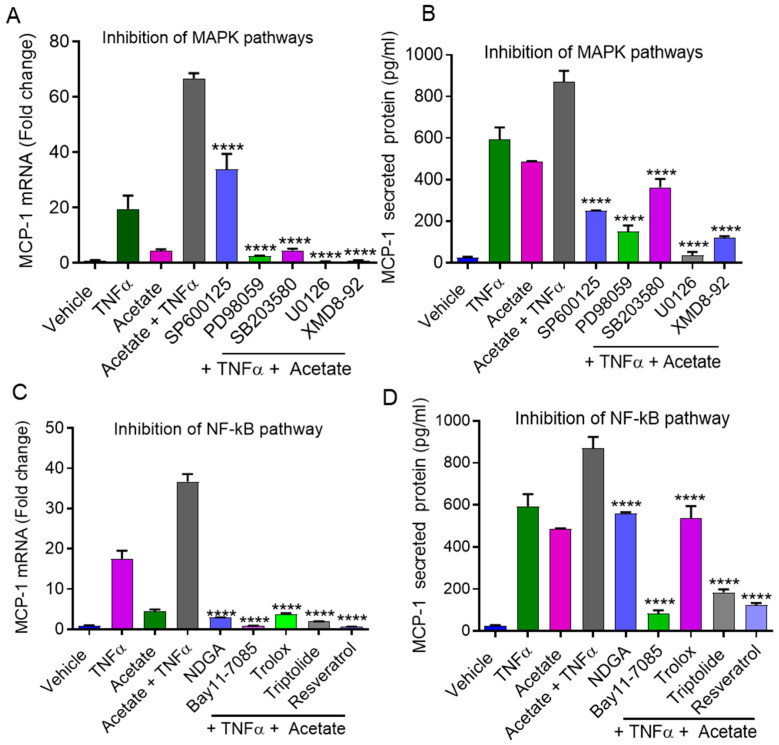
Effect of MAPK and NF-kB pathways inhibitors on MCP-1 induction by acetate/TNFα. THP-1 monocytic cells were pretreated with JNK inhibitor (SP600125: 20 uM) or MEK-ERK inhibitors (PD98059: 10 uM; U0126: 10 uM: XMD-92: 5 uM) or p38 inhibitor (SB203580: 10 uM) for 1 h and then treated with acetate/TNFα for 24 h. Cells and supernatants were collected. (**A**) Cells were used for the isolation of total RNA to assess the MCP-1 gene expression by real-time RT-PCR. (**B**) Secreted levels of MCP-1 protein were determined in supernatants by ELISA. THP-1 cells were pretreated with NF-κB inhibitors (NDGA, 30 uM; Bay11-7085, 10 uM; Trolox, 10 µM; Triptolide, 10 µM; Resveratrol, 15 uM) for 1 h and then treated with acetate (100 mM)/TNFα (10 ng/mL) for 24 h. (**C**,**D**) MCP-1 was determined by RT-PCR and ELISA. The results obtained from three independent experiments are shown. The data are presented as mean ± SEM (n = 3). **** *p* < 0.0001.

**Figure 7 ijms-22-07683-f007:**
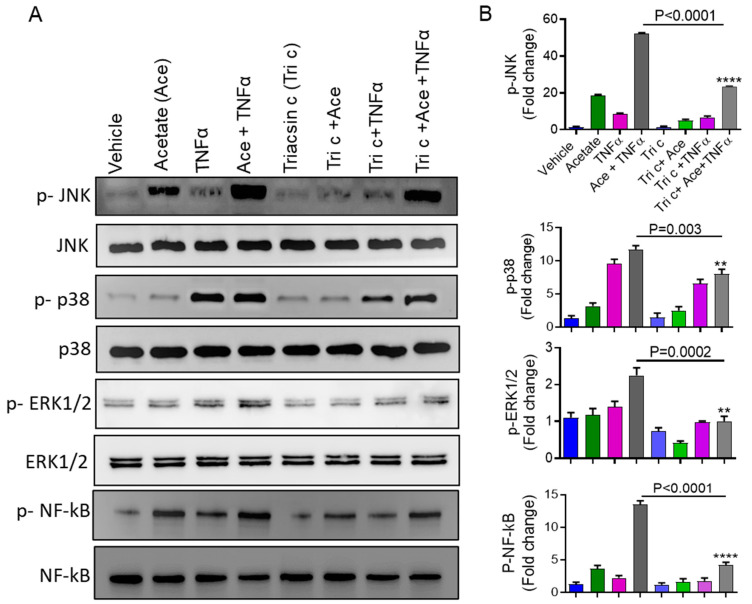
Impaired acetate/TNFα-induced activation of MAPK/NF-κB in triacsin c pretreated cells. (**A**) Cells were preincubated with ACSL1 inhibitor triacsin c (4 uM) for 1 h and subsequently stimulated with acetate or TNFα or acetate/TNFα. The levels of phosphorylated MAPKs and NF-κB were determined by Western blot. The corresponding Western blot for the total protein levels is depicted in the bottom panel. (**B**) Quantification of Western blot. The data are presented as mean ± SEM (n = 3). ** *p* < 0.01, **** *p* < 0.0001.

**Figure 8 ijms-22-07683-f008:**
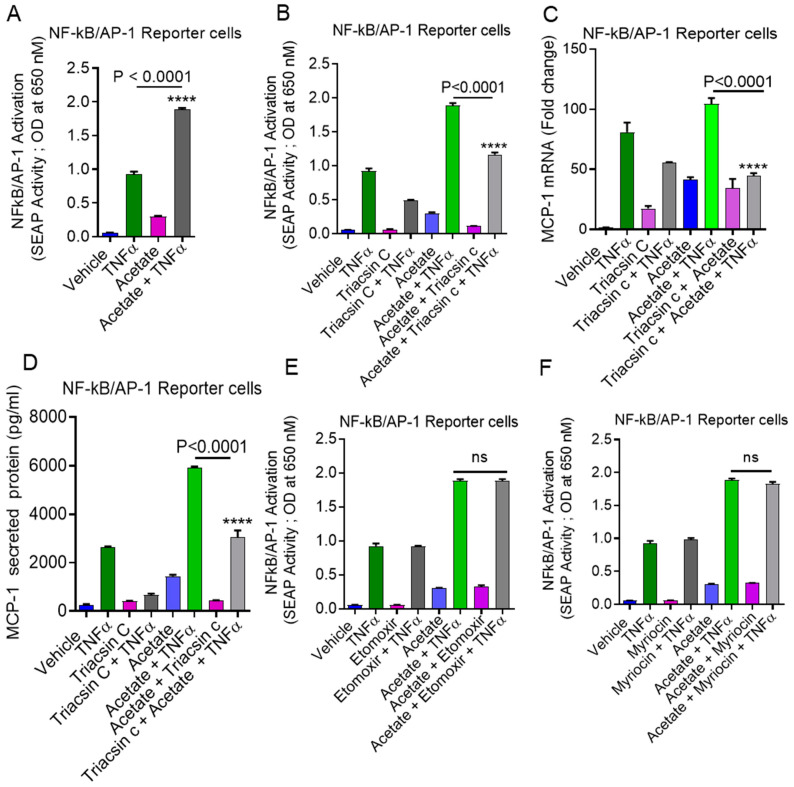
TNFα together with acetate increases NF-κB/AP-1 activity. (**A**) THP1-XBlue cells (monocytic cells stably expressing a SEAP reporter inducible by NF-κB and AP-1) were treated with vehicle or TNFα and acetate alone or in combination, for 24 h. Culture media were collected. Cell culture media were assayed for SEAP reporter activity (the degree of NF-κB/AP-1 activation). (**B**) Cells were pretreated with triacsin c and then exposed to acetate/TNFα. Degree of NF-κB/AP-1 activation was measured. (**C**) Total RNA was isolated from reporter cells to assess the MCP-1 gene expression using real-time RT-PCR. (**D**) MCP-1 protein was determined in the supernatant. (**E**,**F**) Reporter cells were treated with etomoxir or myriocin before being treatedwith acetate/TNFα. Degree of NF-κB/AP-1 activation measured. The results obtained from three independent experiments are shown. All data are expressed as mean ± SEM (n = 3). **** *p* < 0.0001 versus TNFα alone. ns indicates non-significant.

**Figure 9 ijms-22-07683-f009:**
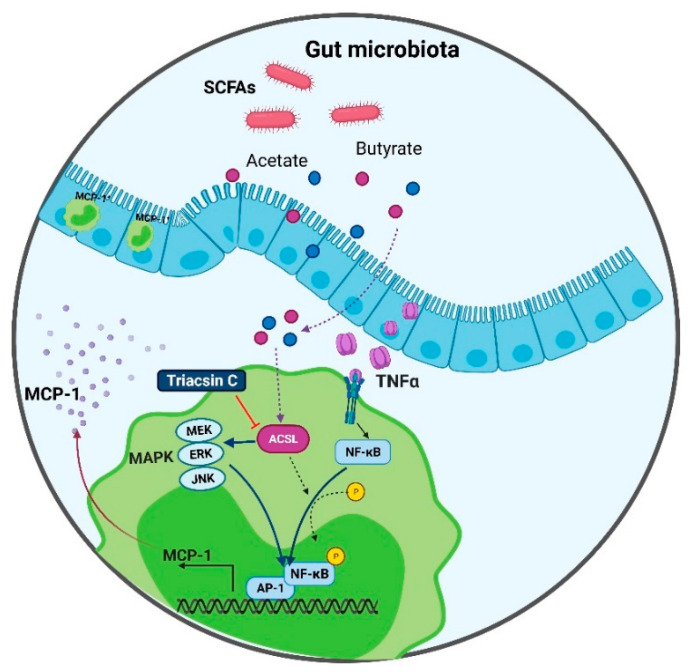
Schematic illustration of signaling pathway underlying the cooperative relationship between acetate and TNF-α for MCP-1 production.

## Data Availability

The data that support the findings of this study are available from the corresponding author upon reasonable request.

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
