# Peer review of "Short Chain Fatty Acid Acetate Increases TNFα-Induced MCP-1 Production in Monocytic Cells via ACSL1/MAPK/NF-κB Axis"

_ijms, 2021, doi:10.3390/ijms22147683_

Round 1

Reviewer 1 Report

In this paper, the authors investigated whether acetate modulates TNFα-mediated production in monocytes/macrophages and the mechanisms underlying this modulation. They proved the proinflammatory effects of acetate on TNF-α-mediated MCP-1 production via the ACSL1/MAPK/NF-κB axis in monocytes, while a paradoxical effect was observed in macrophages.

Since this manuscript it was a lot improved after revision, I consider that it can be accepted for publication.

Author Response

Response to Reviewer 1 Comments

We thank the esteemed reviewer for his encouraging remarks and approval of our work for publication

Comments and Suggestions for Authors

In this paper, the authors investigated whether acetate modulates TNFα-mediated production in monocytes/macrophages and the mechanisms underlying this modulation. They proved the proinflammatory effects of acetate on TNF-α-mediated MCP-1 production via the ACSL1/MAPK/NF-κB axis in monocytes, while a paradoxical effect was observed in macrophages.

Since this manuscript it was a lot improved after revision, I consider that it can be accepted for publication.

We thank the esteemed reviewer for his encouraging remarks and approval of our work for publication.

Submission Date

16 June 2021

Date of this review

18 Jun 2021 11:10:33

Reviewer 2 Report

Major

  The manuscript entitled “Short-chain fatty acid acetate increases TNFα-induced MCP-1 production in monocytic cells via ACSL1/MAPK/NF-κB axis” by Al-Roub et al. showed that the TNFα-mediated increases in MCP-1 expression and secretion were upregulated by SCFAs in THP-1 monocytic cells, but were oppositely downregulated in THP-1 macrophages. The authors moreover described that the upregulation of MCP-1 was mediated through the activation of ACSL1/MAPK/NF-κB axis. This study is interesting but have some problems as follows.

  In data, there is never description of number of experiments. The authors should add n values to all experimental data. In statistics, the multiple comparisons should be used for the significance test in more than three groups.

  The authors do not discuss about involvement of SCFA receptors including FFAR2 and FFAR3. I have found the paper that human monocyte expresses FFA2 and FFA3, and treatment with acetate SCFA or FFAR2- and FFAR3-specific synthetic agonists to the human monocytes display elevated p38 phosphorylation and attenuated C5, CCL1, CCL2, GM-CSF, IL-1α, IL-1β and ICAM-1 inflammatory cytokine expression (Ang Z et al. Scientific Reports 6: 34145). I hope the authors add discussion about the SCFA receptors.

  Concentration of 100 mM of acetate exposing to THP-1 monocytic cells is too high. That concentration might achieve in the large intestinal lumen, but usually improbable in the body, I guess. Please describe the basis of concentrations of SCFAs used.

  In data of Figure 1CD, quantification of Western blot (C) and morphometry analysis (D) should be performed.

Minor

Title: Add hyphen to “TNFα induced MCP-1 production” as “TNFα-induced MCP-1 production”

Abstract, line 25: Change the sentence “Monocytic cells were exposed to acetate with/without TNFα and MCP-1 expression was measured.” to “Monocytic cells were exposed to acetate with/without TNFα for 24h, and MCP-1 expression was measured.”

Author Response

Response to Reviewer 2 Comments

Manuscript ID: ijms-1282253

Title: Short chain fatty acid acetate increases TNFα-induced MCP-1 production in monocytic cells via ACSL1/MAPK/NF-κB axis

We thank the reviewer for his thoughtful comments. Please see below point by point responses to comments.

Comments and Suggestions for Authors

Major

The manuscript entitled “Short-chain fatty acid acetate increases TNFα-induced MCP-1 production in monocytic cells via ACSL1/MAPK/NF-κB axis” by Al-Roub et al. showed that the TNFα-mediated increases in MCP-1 expression and secretion were upregulated by SCFAs in THP-1 monocytic cells, but were oppositely downregulated in THP-1 macrophages. The authors moreover described that the upregulation of MCP-1 was mediated through the activation of ACSL1/MAPK/NF-κB axis. This study is interesting but have some problems as follows.

In data, there is never description of number of experiments. The authors should add n values to all experimental data. In statistics, the multiple comparisons should be used for the significance test in more than three groups.

Response: The results of each experiment were obtained from three independent experiments, and each experiment was performed in triplicate. The data were expressed as mean ± SEM (n= 3); * P < 0.05; ** P < 0.01, *** P < 0.001, **** P < 0.0001,   when compared with TNF-α alone. Furthermore, One-way ANOVA followed by Tukey's test were used to compare means between groups and statistics were accordingly revised in all Figs.

The authors do not discuss about involvement of SCFA receptors including FFAR2 and FFAR3. I have found the paper that human monocyte expresses FFA2 and FFA3, and treatment with acetate SCFA or FFAR2- and FFAR3-specific synthetic agonists to the human monocytes display elevated p38 phosphorylation and attenuated C5, CCL1, CCL2, GM-CSF, IL-1α, IL-1β and ICAM-1 inflammatory cytokine expression (Ang Z et al. Scientific Reports 6: 34145). I hope the authors add discussion about the SCFA receptors.

Response: Thanks for the comment. Accordingly, role of FFAR2 and FFAR3 as SCFAs and their distinct expression of GM-CSF and IL-1α/β in FFAR2/FFAR3 knockout mice in response to acetate and synthetic agonists of FFAR2/3 has been added to the discussion part (Lines 293-303). Reference (Ang Z et al. Scientific Reports 6: 34145) was included.

Concentration of 100 mM of acetate exposing to THP-1 monocytic cells is too high. That concentration might achieve in the large intestinal lumen, but usually improbable in the body, I guess. Please describe the basis of concentrations of SCFAs used.

Response: Thanks for the valuable comment. We used 100 mM sodium acetate for monocyte treatment in vitro, based on the previous studies related to SCFAs ranges found in colonic lumen as mentioned [1, 2]. We also tested 50mM concentration, it also worked.

  1. 1. Wong, J. M.; de Souza, R.; Kendall, C. W.; Emam, A.; Jenkins, D. J., Colonic health: fermentation and short chain fatty acids. J Clin Gastroenterol 2006, 40, (3), 235-43.

2.Li, M.; van Esch, B.; Wagenaar, G. T. M.; Garssen, J.; Folkerts, G.; Henricks, P. A. J., Pro- and anti-inflammatory effects of short chain fatty acids on immune and endothelial cells. Eur J Pharmacol 2018, 831, 52-59.

In data of Figure 1CD, quantification of Western blot (C) and morphometry analysis (D) should be performed.

Response: In regard with the comment, we have done both blot densitometry (revised D) as well as morphometric (revised F) analysis in the revised Fig 1.

Minor

Title: Add hyphen to “TNFα induced MCP-1 production” as “TNFα-induced MCP-1 production

Response: Corrected as suggested.

Abstract, line 25: Change the sentence “Monocytic cells were exposed to acetate with/without TNFα and MCP-1 expression was measured.” to “Monocytic cells were exposed to acetate with/without TNFα for 24h, and MCP-1 expression was measured.”

Response: Modified as suggested

Submission Date

16 June 2021

Date of this review

28 Jun 2021 19:41:07

Round 2

Reviewer 2 Report

The authors sufficiently revised the manuscript, so I recommend editors to accept the revised manuscript.

This manuscript is a resubmission of an earlier submission. The following is a list of the peer review reports and author responses from that submission.

Round 1

Reviewer 1 Report

The authors have presented significant amount of data for proving synergistic effects of acetate and TNFa in mediating MCP-1 secretion from monocytes. Despite all the data, the work is merely additive to the earlier reports on TNFa, monocytes and acetate (including earlier publications from the same group). Besides, there are several other factors that reduce the merit of the paper, for example, discrepancies between text and figures (example, figure 3a); very high concentration of acetate used; no significant effect of triacsin alone.

Author Response

Response to Reviewer 1 Comments

Ref: Revised Submission ID: ijms-1201696

We thank the reviewer for your comments. Please see below point by point responses to his comments.

Comments and Suggestions for Authors

The authors have presented significant amount of data for proving synergistic effects of acetate and TNFa in mediating MCP-1 secretion from monocytes. Despite all the data, the work is merely additive to the earlier reports on TNFa, monocytes and acetate (including earlier publications from the same group). Besides, there are several other factors that reduce the merit of the paper, for example, discrepancies between text and figures (example, figure 3a); very high concentration of acetate used; no significant effect of triacsin alone.

Authors’ response: In this study, we described for the first time that acetate increases TNFα mediated MCP-1 production in monocytic cells. We also found that ACSL1, a key enzyme of lipid/fatty acids metabolism, plays a major role in the synergistic production of MCP-1 under the influence of acetate/TNFα. Figure 3a which was from monocytic cells and mistakenly placed under macrophages and now replaced with correct figure related to macrophages.

Reviewer 2 Report

The authors found that ACSL1 plays a novel role in acetic acid/TNFα-mediated MCP-1 production in monocytic cells, dependent on activation of MAPK and NF-κB signaling pathways. These results suggest an association between acetate concentration, TNFα expression, and ACSL1 activity in obesity, which may be a potential cause of metabolic inflammation.

Overall, the paper seems to be well-written and interesting. I think this paper could potentially be suitable for publication in International Journal of Molecular Sciences, but there are several improvements that should be made.

  1. Regarding ACSL1 (long-chain fatty acid acyl-CoA synthase 1), could you explain in more detail which molecule this enzyme acts on to cause the phenomenon revealed in this study? Or can we assume that long-chain acyl-CoAs, the product of ACSL1, have a direct effect on each molecule?

  1. Figure 1C: “B-actin” should be “b-actin”.

  1. Figure 3: You mentioned that TNFα mediated MCP-1 production was significantly downregulated in macrophages pretreated with acetate (Figure 3A), butyrate (Figure 3B), or propionate (Figure 3C). However, TNFα mediated MCP-1 production was upregulated in macrophages pretreated with acetate shown in Figure 3A. Which is correct?

  1. Figure 4: “Inhibition of Acyl CoA synthetase (ACSL1)” should be “Inhibition of acyl-CoA synthetase (ACSL1)”.

  1. Figures 5A and 5B: “(Fold change )” should be “(Fold change)”.

  1. Figure 7: “p-NF-kB” and “NF-kB” should be “p-NF-kB” and “NF-kB”, respectively.

  1. Figures 8B, 8E and 8F: “(SEAP Activity; ODat 650 nM)” should be “(SEAP Activity; OD at 650 nM)”.

  1. Figure 8C: “(Fold change )” should be (Fold change).

  1. Figure 9: Please increase the resolution of the diagrams and the size of the text so that the text can be read clearly. “TNF-a” should be “TNFa”. What is the difference between MCP-1+ and MCP-1?

  1. Line 20: “MCP-1” should be “Monocyte chemoattractant protein-1 (MCP-1, also known as Chemokine (C-C motif) ligand 2 (CCL2))”.

  1. Line 59: “SCFAs” should be “Short-chain fatty acids (SCFAs)”.

  1. Line 74: “CCR2” should be “chemokine (C-C motif) receptor 2 (CCR2)”.

  1. Line 95 and others: “acetate (100mM) and TNFα (10ng)” should be “acetate (100 mM) and TNFα (10 ng)”. Please put a space between the number and the unit. There are many places in the text that should be corrected. “24h” should be “24 h”.

  1. Line 109: “butyrate(2mM), propionate(10mM)” should be “butyrate (2 mM), propionate (10 mM)”.

  1. Lines 110 and 121: “24h” should be “24 h”.

  1. Line 117: “(MCP-1 p = 0.0001)” should be “(p = 0.0001)”.

  1. Lines 131-132: “etomoxir (carnitine palmitoyltransferase I) or myriocin (serine palmitoyltransferase)” should be “etomoxir (carnitine palmitoyltransferase I (CPT1)) or myriocin (serine palmitoyltransferase (SPT))”.

  1. Line 135: “Triacsin c:(4uM), etomoxir (10uM), myriocin (50nM)” should be “Triacsin c (4 mM), etomoxir (10 mM), myriocin (50 mM)”. There are many other similar parts in the text, so please correct nM to m

  1. Line 249: “short-chain fatty acids (SCFAs)” should be “SCFAs”.

  1. Line 290 and others: “Tnfa” should be “Tnfa”. A lot of “Tnfa” can be found in the manuscript.

  1. Line 330: Please put the appropriate reference number in “(REF)”.

  1. Line 332: “1×106 cells” should be “1×106 cells”.

  1. Lines 333-334: “Na Acetate (100 mM; Sigma, MO, USA), Na Butyrate 333 (2 mM; Sigma, MO, USA), Na propionate (10 mM; Sigma, MO, USA)” should be “sodium acetate (100 mM; Sigma, MO, USA), sodium butyrate 333 (2 mM; Sigma, MO, USA), sodium propionate (10 mM; Sigma, MO, USA).

  1. Line 340: “1×106 cells” should be “1×106 cells”.

  1. Line 357: Please put the appropriate reference number in “(REF)”.

  1. Line 366: “Small interfering RNA (siRNA) transfection” should be “5. Small interfering RNA (siRNA) transfection”. There should be a space between lines 365 and 366.

  1. Line 367: Please correct the units, “100 l”.

  1. Line 373: “36 hours” and “24 hours” should be “36 h” and “24 h”, respectively.

  1. Line 377: “5. Immunocytofluorescence” should be “4.6. Immunocytofluorescence”.

  1. Line 379: “1×106 monocytic cells” should be “1×106 monocytic cells”.

  1. Line 393: “6. Western blot analysis” should be “4.7. Western blot analysis”.

  1. Lines 394-395: “30 minutes” should be “30 min”.

  1. Line 398: “1 hour” should be “1 h”.

  1. Line 402: “B-actin” should be “b-actin”.

  1. Line 403: “2 hours” should be “2 h”.

  1. Line 408: “7. Measurement of NF-κB/AP-1 activity” should be “4.8. Measurement of NF-κB/AP-1 activity

  1. Line 414: “24h” should be “24 h”.

  1. Line 417: “8. Statistical analysis” should be “4.9. Statistical analysis”.

  1. Lines 421-422: There should be a space between lines 421 and 422.

  1. Author Contributions: I don't think everyone's role is listed. Could you please check it out?

  1. References: Please align the style of References correctly. For example, regarding journal names, some are abbreviated, some are not.

Author Response

Response to Reviewer 2 Comments

Ref: Revised Submission ID: ijms-1201696

We thank the reviewer for his thoughtful comments. Please see below point by point responses to his comments.

Comments and Suggestions for Authors

The authors found that ACSL1 plays a novel role in acetic acid/TNFα-mediated MCP-1 production in monocytic cells, dependent on activation of MAPK and NF-κB signaling pathways. These results suggest an association between acetate concentration, TNFα expression, and ACSL1 activity in obesity, which may be a potential cause of metabolic inflammation.

Overall, the paper seems to be well-written and interesting. I think this paper could potentially be suitable for publication in International Journal of Molecular Sciences, but there are several improvements that should be made.

  1. Regarding ACSL1 (long-chain fatty acid acyl-CoA synthase 1), could you explain in more detail which molecule this enzyme acts on to cause the phenomenon revealed in this study? Or can we assume that long-chain acyl-CoAs, the product of ACSL1, have a direct effect on each molecule?

Authors’ response: ACSL1, a member of rate-limiting enzymes in fatty acid metabolism, catalyze the bioconversion of exogenous or de novo synthesized fatty acids to their corresponding fatty acyl-CoAs. We assumed a direct involvement of acyl-CoAs in the synergistic production of MCP-1 by acetate and TNFα in monocytic cells.

  1. Figure 1C: “B-actin” should be “b-actin”.

             Authors’ response: Done as suggested.

  1. Figure 3: You mentioned that TNFα mediated MCP-1 production was significantly downregulated in macrophages pretreated with acetate (Figure 3A), butyrate (Figure 3B), or propionate (Figure 3C). However, TNFα mediated MCP-1 production was upregulated in macrophages pretreated with acetate shown in Figure 3A. Which is correct?

Authors’ response: Thank you for pointing out the error.  Figure 3a which was from monocytic cells and mistakenly placed under macrophages and now it was replaced with correct figure related to macrophages.

  1. Figure 4: “Inhibition of Acyl CoA synthetase (ACSL1)” should be “Inhibition of acyl-CoA synthetase (ACSL1)”.

Authors’ response: Done as suggested.

  1. Figures 5A and 5B: “(Fold change )” should be “(Fold change)”.

            Authors’ response: Done.

  1. Figure 7: “p-NF-kB” and “NF-kB” should be “p-NF-kB” and “NF-kB”, respectively.

Authors’ response: Done.

  1. Figures 8B, 8E and 8F: “(SEAP Activity; ODat 650 nM)” should be “(SEAP Activity; OD at 650 nM)”.

            Authors’ response: Corrected.

  1. Figure 8C: “(Fold change )” should be (Fold change).

      Authors’ response: Corrected

  1. Figure 9: Please increase the resolution of the diagrams and the size of the text so that the text can be read clearly. “TNF-a” should be “TNFa”. What is the difference between MCP-1+ and MCP-1?

Authors’ response:  MAP-1 is a secretary protein and MCP-1+ represents the monocytes with high expression of MCP-1. As per suggested, we included the description in figure legend.

  1. Line 20: “MCP-1” should be “Monocyte chemoattractant protein-1 (MCP-1, also known as Chemokine (C-C motif) ligand 2 (CCL2))”.

 Authors’ response:  Done as suggested.

  1. Line 59: “SCFAs” should be “Short-chain fatty acids (SCFAs)”.

 Authors’ response:  Done as suggested.

  1. Line 74: “CCR2” should be “chemokine (C-C motif) receptor 2 (CCR2)”.

 Authors’ response:  Done as suggested.

  1. Line 95 and others: “acetate (100mM) and TNFα (10ng)” should be “acetate (100 mM) and TNFα (10 ng)”. Please put a space between the number and the unit. There are many places in the text that should be corrected. “24h” should be “24 h”.

 Authors’ response:  All corrections have been made.

  1. Line 109: “butyrate(2mM), propionate(10mM)” should be “butyrate (2 mM), propionate (10 mM)”.

 Authors’ response:  All corrections have been made.

  1. Lines 110 and 121: “24h” should be “24 h”.

            Authors’ response:  All corrections have been made.

  1. Line 117: “(MCP-1 p = 0.0001)” should be “(p = 0.0001)”.

            Authors’ response:  Done.

  1. Lines 131-132: “etomoxir (carnitine palmitoyltransferase I) or myriocin (serine palmitoyltransferase)” should be “etomoxir (carnitine palmitoyltransferase I (CPT1)) or myriocin (serine palmitoyltransferase (SPT))”.

 Authors’ response:  All corrections have been made.

  1. Line 135: “Triacsin c:(4uM), etomoxir (10uM), myriocin (50nM)” should be “Triacsin c (4 mM), etomoxir (10 mM), myriocin (50 mM)”. There are many other similar parts in the text, so please correct nM to m

 Authors’ response:  All corrections have been made.

  1. Line 249: “short-chain fatty acids (SCFAs)” should be “SCFAs”.

 Authors’ response:  Done.

  1. Line 290 and others: “Tnfa” should be “Tnfa”. A lot of “Tnfa” can be found in the manuscript.

 Authors’ response:  All corrections have been made.

  1. Line 330: Please put the appropriate reference number in “(REF)”.

      Authors’ response:  Reference has been included.

  1. Line 332: “1×106 cells” should be “1×106 cells”.

Authors’ response:  All corrections have been made.

  1. Lines 333-334: “Na Acetate (100 mM; Sigma, MO, USA), Na Butyrate 333 (2 mM; Sigma, MO, USA), Na propionate (10 mM; Sigma, MO, USA)” should be “sodium acetate (100 mM; Sigma, MO, USA), sodium butyrate 333 (2 mM; Sigma, MO, USA), sodium propionate (10 mM; Sigma, MO, USA).

       Authors’ response:  All corrections have been made.

  1. Line 340: “1×106 cells” should be “1×106 cells”.

      Authors’ response:  Corrected.

  1. Line 357: Please put the appropriate reference number in “(REF)”.

      Authors’ response:  Reference was included.

  1. Line 366: “Small interfering RNA (siRNA) transfection” should be “5. Small interfering RNA (siRNA) transfection”. There should be a space between lines 365 and 366.

     Authors’ response:  Correction was made.

  1. Line 367: Please correct the units, “100 l”.

     Authors’ response:  Corrected.

  1. Line 373: “36 hours” and “24 hours” should be “36 h” and “24 h”, respectively.

     Authors’ response:  All corrections have been made.

  1. Line 377: “5. Immunocytofluorescence” should be “4.6. Immunocytofluorescence”.

      Authors’ response:  All corrections have been made.

  1. Line 379: “1×106 monocytic cells” should be “1×106 monocytic cells”.

      Authors’ response:  Corrected.

  1. Line 393: “6. Western blot analysis” should be “4.7. Western blot analysis”.

Authors’ response:  Corrected.

  1. Lines 394-395: “30 minutes” should be “30 min”.

       Authors’ response:  Corrected.

  1. Line 398: “1 hour” should be “1 h”.

       Authors’ response:  Corrected.

  1. Line 402: “B-actin” should be “b-actin”.

       Authors’ response:  Corrected.

  1. Line 403: “2 hours” should be “2 h”.

       .Authors’ response:  Corrected

  1. Line 408: “7. Measurement of NF-κB/AP-1 activity” should be “4.8. Measurement of NF-κB/AP-1 activity

 Authors’ response:  Done.

  1. Line 414: “24h” should be “24 h”.

      Authors’ response:  Corrected.

  1. Line 417: “8. Statistical analysis” should be “4.9. Statistical analysis”.

      Authors’ response:  Done.

  1. Lines 421-422: There should be a space between lines 421 and 422.

      Authors’ response:  Done.

  1. Author Contributions: I don't think everyone's role is listed. Could you please check it out?

       Authors’ response:  Contributions of all authors included.

  1. References: Please align the style of References correctly. For example, regarding journal names, some are abbreviated, some are not.

Authors’ response:  Corrected

Reviewer 3 Report

The authors showed the proinflammatory effects of acetate on TNF-α-mediated MCP-1 production. Additionally, the authors demonstrated that these effects are dependent on the ACSL1/MAPK/NF-33 κB axis in monocytes.

 I consider that the manuscript is valuable and it can be accepted after minor text revision.

Minor comments:

  • Figure 6 “Resveratrol” instead of “Resveratol”
  • Row 124: ACSL1 “is” a key enzyme
  • Row 214: “The” degree of NF-κB/AP1 activation
  • “Short-chain fatty acids (SCFAs) ” should be at Row 59 instead of Row 249
  • “Small interfering RNA (siRNA) transfection” should be the chapter “4.5.” and the next chapters should be numbered accordingly.
  • Row 402: Anti-MCP1 antibody was bought from abcam
  • Delete Rows 424-425

Author Response

Response to Reviewer 3 Comments

Ref: Revised Submission ID: ijms-1201696

We thank the reviewer for his thoughtful comments. Please see below point by point responses to his comments.

Comments and Suggestions for Authors

The authors showed the proinflammatory effects of acetate on TNF-α-mediated MCP-1 production. Additionally, the authors demonstrated that these effects are dependent on the ACSL1/MAPK/NF-33 κB axis in monocytes.

 I consider that the manuscript is valuable, and it can be accepted after minor text revision.

Minor comments:

  • Figure 6 “Resveratrol” instead of “Resveratol”

Authors’ response:  Corrected.

  • Row 124: ACSL1 “is” a key enzyme

Authors’ response: Done.

  • Row 214: “The” degree of NF-κB/AP1 activation

Authors’ response: Done.

  • “Short-chain fatty acids (SCFAs) ” should be at Row 59 instead of Row 249

Authors’ response: Done.

  • “Small interfering RNA (siRNA) transfection” should be the chapter “4.5.” and the next chapters should be numbered accordingly.

Authors’ response: Corrected.

  • Row 402: Anti-MCP1 antibody was bought from abcam

Authors’ response: Corrected.

  • Delete Rows 424-425

Authors’ response: Rows 424-425 were deleted.

Round 2

Reviewer 1 Report

Authors have considerably revised the manuscript with regards to presentation, language and other errors noted before. There are few concerns now noted:

Authors make a case of adipose dysfunction and increase in MCP1 production besides other inflammatory mediators. How do you define this adipose dysfunction (lines 70/71, introduction)? Are the blood monocytes getting affected in the manner as described by the results? How then adipose fits in here, because macrophages, in this case, appear to be refractory to acetate or other SCFAs and TNFa co treatment? From the final schematic it appears that a giant monocytic cell is fragmenting into smaller MCP+ cells. It should be modified. Besides, there are several other discrepancies noted. Acetate does not increase mRNA expression of MCP1 to that extent but increases protein secretion similar to TNFa. How ? Was any TLR agonist used ? SCFAs do not activate TLRs but can augment the signaling downstream of TLR activation by its agonists. Discussion does not highlight the relevance of the work.

Reviewer 2 Report

I thank the authors’ polite reply to my comments.

I have still a few comments on the revised MS. I hope these are helpful.

  1. The authors assumed a direct involvement of acyl-CoAs catalyzed by ACSL1 in the synergistic production of MCP-1 by acetate and TNFα in monocytic cells. If that's the case, it would be better to mention what specific acyl-CoAs are involved in MCP-1 production.

  1. For Figure 9, the resolution has been greatly increased, making it easier to understand. Thank you very much. What are NEMO, IKKa and IKKb in Figure 9, respectively? They do not appear in the text. You need an explanation. Also, I think that the products of ACSL1 (namely acyl-CoAs) are involved in MAPK pathway, etc. rather than ACSL1 itself.